# Navigating the Intersection: Sarcopenia and Sarcopenic Obesity in Inflammatory Bowel Disease

**DOI:** 10.3390/biomedicines12061218

**Published:** 2024-05-30

**Authors:** Valentin Calvez, Guia Becherucci, Carlo Covello, Giulia Piccirilli, Irene Mignini, Giorgio Esposto, Lucrezia Laterza, Maria Elena Ainora, Franco Scaldaferri, Antonio Gasbarrini, Maria Assunta Zocco

**Affiliations:** CEMAD Digestive Disease Center, Fondazione Policlinico Universitario “A. Gemelli” IRCCS, Catholic University of Rome, 00168 Rome, Italy; valentino.calvez@gmail.com (V.C.); guia.becherucci01@icatt.it (G.B.); covellocarlo@gmail.com (C.C.); giulia.piccirilli94@gmail.com (G.P.); irene.mignini@guest.policlinicogemelli.it (I.M.); giorgio.esposto2@gmail.com (G.E.); lucrezia.laterza@policlinicogemelli.it (L.L.); mariaelena.ainora@policlinicogemelli.it (M.E.A.); franco.scaldaferri@policlinicogemelli.it (F.S.); antonio.gasbarrini@unicatt.it (A.G.)

**Keywords:** inflammatory bowel disease, sarcopenia, sarcopenic obesity, malnutrition, inflammation, muscle mass, gut microbiota

## Abstract

Inflammatory bowel diseases (IBDs) are intricate systemic conditions that can extend beyond the gastrointestinal tract through both direct and indirect mechanisms. Sarcopenia, characterized by a reduction in muscle mass and strength, often emerges as a consequence of the clinical course of IBDs. Indeed, sarcopenia exhibits a high prevalence in Crohn’s disease (52%) and ulcerative colitis (37%). While computed tomography and magnetic resonance imaging remain gold-standard methods for assessing muscle mass, ultrasound is gaining traction as a reliable, cost-effective, and widely available diagnostic method. Muscle strength serves as a key indicator of muscle function, with grip strength test emerging nowadays as the most reliable assessment method. In IBDs, sarcopenia may arise from factors such as inflammation, malnutrition, and gut dysbiosis, leading to the formulation of the ‘gut–muscle axis’ hypothesis. This condition determines an increased need for surgery with poorer post-surgical outcomes and a reduced response to biological treatments. Sarcopenia and its consequences lead to reduced quality of life (QoL), in addition to the already impaired QoL. Of emerging concern is sarcopenic obesity in IBDs, a challenging condition whose pathogenesis and management are still poorly understood. Resistance exercise and nutritional interventions, particularly those aimed at augmenting protein intake, have demonstrated efficacy in addressing sarcopenia in IBDs. Furthermore, anti-TNF biological therapies showed interesting outcomes in managing this condition. This review seeks to furnish a comprehensive overview of sarcopenia in IBDs, elucidating diagnostic methodologies, pathophysiological mechanisms, and clinical implications and management. Attention will also be paid to sarcopenic obesity, exploring the pathophysiology and possible treatment modalities of this condition.

## 1. Introduction

The term “sarcopenia”, introduced by Rosenberg in 1988 to describe the loss of muscle mass in older adults, has been subject to various definitions [1]. Recently, sarcopenia has been classified as a condition marked by a significant reduction in muscle mass, along with a decrease in muscle strength, where poor physical performance is often considered as an indicator of severe sarcopenia [2].

The prevalence of sarcopenia in the general population is currently highly variable, ranging from 0.2% to 86.5% due to the use of different classification systems and cut-offs [2]. According to the latest revision of the European consensus on sarcopenia (EWGSOP2), it is more frequent in men than in women, with a prevalence of 11% and 2%, respectively [3].

It is a challenging condition that can affect people of all ages; however age-related factors are often the primary cause of sarcopenia, leading to a loss of lean mass and reduced functionality. Secondary sarcopenia can be caused by various factors [3]. Among these, inflammatory bowel diseases (IBDs)—which mainly include Crohn’s disease (CD) and ulcerative colitis (UC)—represent a high-risk factor for the development of sarcopenia as they are characterized by chronic inflammation, malabsorption, and increased basal energy expenditure due to an inflammatory state, with significant alterations in lipid and carbohydrate metabolism and altered dietary intake. A recent meta-analysis by Ryan et al., including 658 IBD patients, found a prevalence of sarcopenia in 52% of patients with CD and 37% with UC [4]. Therefore, it is crucial to address these underlying causes and take proactive steps to prevent or manage sarcopenia.

It is well established that individuals diagnosed with inflammatory bowel disease (IBD) are at a greater risk of developing sarcopenia, a condition that can have detrimental effects on clinical outcomes. In fact, recent research has demonstrated that the presence of sarcopenia in IBD patients significantly increases the likelihood of requiring surgery and experiencing postoperative complications [5] such as infections and blood clots, which can lead to prolonged hospitalization [6,7,8].

Moreover, sarcopenia has been associated with an increased risk of re-hospitalization, an impaired quality of life (QoL), increased healthcare costs [8], and poor response to therapy [9].

It is noteworthy that sarcopenia is a critical factor that negatively impacts biological therapy. Specifically, it represents an adverse prognostic factor for the achievement of endoscopic remission and has also been found to be associated with primary loss of response [10]. As such, patients with sarcopenia may experience inferior outcomes in response to biological therapy. Given the importance of achieving positive treatment outcomes, healthcare professionals and researchers must be aware of the negative impact of sarcopenia when considering treatment options for patients with inflammatory bowel disease.

It is now widely recognized that sarcopenia is not limited to individuals with low body weight. According to a recent study by Wei et al. 2023, it often affects normal or overweight patients and can even lead to the development of sarcopenic obesity [11]. This condition requires particular attention in the management of IBD patients as 15% to 40% of individuals with IBD are obese [12,13,14,15], and it is often associated with greater adverse complications compared to patients exclusively affected by either obesity or sarcopenia [16]. A comprehensive assessment of sarcopenia should therefore be conducted in all IBD patients to evaluate body composition, muscle function, and physical performance [3].

This review aims to provide a comprehensive summary of the current understanding of sarcopenia in individuals with IBD. The focus will be on the diagnostic methods employed, the pathophysiological mechanisms underlying sarcopenia in IBD patients, and the associated clinical outcomes. Additionally, we will review the different treatment approaches utilized to manage sarcopenia in individuals with IBD. 

Lastly, in light of the emerging issue of sarcopenic obesity in IBD, we will direct our attention to this relatively understudied entity, exploring its implications and potential treatments.

## 2. Diagnostic Methods to Identify Sarcopenia in IBD Patients

In 2018, the European Working Group on Sarcopenia in the Elderly (EWGSOP2) came up with an updated agreement applicable to sarcopenic patients of all ages according to which the diagnosis of sarcopenia is determined by the assessment of three key factors: muscle mass, muscle strength, and physical performance [3].

### 2.1. Muscle Mass Assessment

Currently, several methods are available for assessing muscle mass, each with its strengths and limitations. While magnetic resonance imaging (MRI) and computed tomography (CT) are considered the gold standards, they have certain drawbacks that limit their use [17,18]. Dual-energy X-ray absorptiometry (DXA) and bioimpedance analysis (BIA) are also reliable and cost-effective techniques for evaluating skeletal muscle mass [19,20]. Additionally, ultrasound is gaining popularity as a reliable method for diagnosis [21]. 

Computed tomography is increasingly used to analyze sarcopenia due to its high resolution and ability to distinguish muscle from surrounding tissue. It can assess both the quantity and quality of muscles by estimating the total muscle mass and calculating the visceral adipose and subcutaneous tissues, thereby producing an estimate of the body composition. The validity of the parameters revealed by CT on body composition has been verified in IBD patients by numerous studies [22,23,24,25]. A recent retrospective multicenter study of 187 IBD patients compared various parameters that could be obtained from cross-sectional CT images at the third lumbar vertebrae (L3) level. Results revealed that muscle volume was strongly associated with the short-term prognosis of prolonged hospital stay. Furthermore, muscle volume and visceral adipose tissue volume were found to be correlated with the long-term prognosis of bowel resection [26]. Importantly, CT has inherent limitations, including exposing patients to harmful ionizing radiation and high economic costs. However, IBD patients often undergo CT scans to monitor disease progression, meaning that a single examination could provide significant information regarding IBD and body composition without posing additional risks to the patient.

Magnetic resonance imaging has emerged as a reliable method for assessing muscle quantity and quality abnormalities, including muscle injury, edema, myosteatosis, or myofibrosis [27,28]. It can even assess muscle elasticity and contraction, making it an optimal method for estimating body composition and assessing sarcopenia [29]. In a recent study, 35 CD patients were evaluated using routine MRI enterography to measure skeletal muscle signal intensity. The study found that patients with increased muscle fat content, and therefore lower skeletal muscle intensity, underwent bowel resection sooner. These findings suggest that enteroRM is a reliable tool to predict unfavorable disease outcomes and has good reproducibility in CD patients [30]. An RCT of CD patients undergoing surgical resection analyzed CT and MRI images taken in the 12 months prior to enrolment showing that excessive visceral adiposity is associated with worse outcomes in CD patients [31]. MRI is safer than CT as it does not expose patients to radiation. However, it remains an expensive imaging technique with limited availability compared to other methods [32].

Bioelectrical impedance analysis (BIA) is a valid, convenient, and non-invasive tool to indirectly assess body composition and works by applying a weak electric current that passes through the body and encounters different levels of resistance depending on the tissue involved. Parameters like reactance, resistance, and phase angle are combined with anthropometric variables (weight and height), sex, and age to analyze body composition [33,34]. A recent study conducted on IBD patients has found that phase angle can be a useful nutritional indicator to predict disease severity and response to treatment. However, inflammation can affect the phase angle in IBD patients, leading to inaccurate results. Nevertheless, the reliability of BIA in assessing body composition in IBD patients has been shown [35] and was further confirmed by a recent retrospective study of 139 IBD patients, which additionally described that the results varied depending on the disease’s activity status. Specifically, patients with inactive disease had significantly higher values for body moisture, muscle mass, skeletal muscle mass, body mass index (BMI), and minerals [36].

BIA is convenient, cost-effective, fast, and safe to use but is considered less reliable than other methods and assumes fixed tissue hydration, which may not apply to critically ill patients. As a result, interpretation can be more challenging [20].

Dual-energy X-ray absorptiometry (DXA), the gold standard for diagnosing and monitoring osteopenia and osteoporosis, uses a dual-energy beam generated by an X-ray source that attenuates as it passes through the body, influenced by energy intensity and tissue thickness and density [37,38]. It can also quantify fat mass, distinguishing it into visceral adipose tissue (VAT), subcutaneous adipose tissue (SAT), and bone mineral content. In a recent prospective study, the use of DXA was evaluated as a valid tool for analyzing body composition in IBD patients. The study also noted an improvement in skeletal muscle index upon resolution of inflammation [39]. DXA is an excellent method for analyzing body composition [21], but many physicians are reluctant to accept it as the gold standard due to its limitations, which include radiation exposure, albeit slight, and the need for constant hydration of lean soft tissue, which is also a variable factor [38,40].

Ultrasonography imaging is a further safe, precise, widely available and easy-to-use method, which allows for a quick and accurate estimation of quality and quantity of individual muscle groups [41]. Quantity can be assessed using parameters such as muscle volume (MV), CSA, and muscle thickness (MT), whereas echo intensity (EI), pennation angle (PA), fascicle length (FL), and physiological cross-sectional area (PCSA) qualitatively describe the muscle [42,43,44]. EI reflects the adipose and fibrous tissue within the muscle and thus allows assessment of possible conditions of myosteatosis and myofibrosis, often associated with sarcopenia [45]. Mulinacci et al., in a prospective study of 153 IBD patients, showed that muscle ultrasound has good diagnostic accuracy in detecting sarcopenia compared to BIA [46]. Nguyen et al. conducted a study to assess muscularity using various techniques in both IBD patients and healthy controls. They found that ultrasound was the most accurate method for measuring muscle mass compared to BIA and DXA. Furthermore, they discovered that ultrasound had the highest level of agreement with the DXA-derived muscle mass index compared to BIA [47]. Ultrasound is indeed gaining popularity in the assessment of sarcopenia. However, there is no full agreement on standardized methods, and universal reference cut-offs are still lacking [21], especially in IBD patients.

In summary, various diagnostic methods are available to assess skeletal muscle, which are summarized in Table 1. DXA and BIA offer quantitative assessments, while CT and MRI provide both qualitative and quantitative measures of muscle. However, these tools have limitations for clinical practice, particularly due to radiation exposure and high costs. Ultrasound, on the other hand, is a very safe method that can provide precise qualitative and quantitative assessments of muscle. It is closely related to MRI [48,49], CT [50], and DXA [51], thus offering several advantages over other methods. 

### 2.2. Muscle Strength Assessment 

Muscle strength is currently the most reliable indicator of muscle function and is recognized as the main parameter for conducting an initial assessment of sarcopenia [3]. This parameter is mainly investigated with the grip strength test, which is a reliable indicator of the maximum force generated by the extrinsic and intrinsic muscles of the hand [52,53,54]. This test is useful in identifying the risk of sarcopenia in normal weight or overweight patients [55,56]. A recent study found that individuals with IBD exhibit lower grip strength compared to healthy individuals [57]. However, this test can be influenced by inflammation, corticosteroids, hypoxia, electrolyte imbalances, and oxidative stress [58] which could limit its usefulness in IBD patients. 

### 2.3. Physical Performance Assessment 

Sarcopenia can impact physical performance, which is a measure of whole-body function related to movement. It is important to note that physical performance tests are not commonly used for patients with inflammatory bowel disease (IBD). These physical performance tests are only effective in severe cases of sarcopenia [18,59]; in mild or moderate cases of sarcopenia, these tests seem to be less effective, as physical performance, such as the ability to sit on a chair, is generally not significantly affected [59].

### 2.4. Malnutrition Risk Screening

Malnutrition is one of the contributing factors to the development of sarcopenia. In this regard, nutritional screening, using questionnaires and anamnesis, is useful to make an initial assessment of the risk of malnutrition to allow early intervention and prevent its progression. Currently, there are several questionnaires available. Some of these are specifically designed for individuals with inflammatory bowel disease (IBD), such as the IBD-NST and SaskIBD-NR questionnaires. These tools have been shown to be effective in identifying factors that may lead to malnutrition in IBD [60]; however, markers of inflammation, which significantly influence nutritional status, are not included. The Malnutrition Inflammation Risk Tool (MIRT) was the first questionnaire to incorporate CRP scores and is now one of the most recommended tools for these patients [61]. MIRT can be helpful and convenient, but it has limitations as it heavily relies on BMI.

Indeed, these screening tools are primarily based on weight or BMI indices, which may not always accurately indicate the presence of sarcopenia. Therefore, it can be inferred that these questionnaires are not entirely reliable and may not always provide accurate results [60].

## 3. Pathogenetic Mechanisms of Sarcopenia in IBD

The hypothesis of a “gut–muscle axis” has recently gained traction in the scientific community. According to this hypothesis, sarcopenia is influenced by a combination of inflammation, gut dysbiosis, and malnutrition, and their intricate relationship appears to play a significant role in its pathogenesis [62,63]. Indeed, a premature and “accelerated” development of sarcopenia has been identified in groups which share common features such as chronic inflammation, malnutrition, and immobility, IBD being one of them [4].

### 3.1. Primary Sarcopenia

When discussing muscle quantity and quality, it is important to acknowledge that sarcopenia can be categorized into two types: primary sarcopenia, attributed to aging, and secondary sarcopenia, caused by other factors, including IBD [3]. However, primary and secondary sarcopenia may overlap and share some pathogenic mechanisms. Moreover, the mechanisms of primary sarcopenia may have an acceleration in IBD.

As individuals age, the balance between anabolic and catabolic muscle processes deteriorates, leading to a gradual decline in muscle cells resulting in a generalized loss of muscle, muscle strength, and physical performance [64]. These alterations are attributable to the significant reduction in motor neurons innervating muscle fibers and to the deterioration of skeletal muscle cells. Multiple factors are involved in these processes, including diminished mitochondrial function, changes in gene expression, decreased insulin sensitivity, and altered neuromuscular signaling [65,66,67]. 

With age, the secretion of several hormones that impact skeletal muscle also declines, including testosterone, estradiol, growth hormone (GH), and insulin-like growth factor (IGF1). This decrease in anabolic hormones occurs concurrently with an increase in catabolic hormones like cortisol and angiotensin II, which may contribute to age-related muscle atrophy [68]. Studies have shown that the physiology of muscles can be affected by additional factors, including thyroid hormones, vitamin D, and adipokines [69,70]. Upon these events, a reduction in both the quantity and scale of type II muscle fibers is observed, and this, in turn, leads to the promotion of intramuscular and intermuscular adipose tissues [66,67]. Concomitant with the process of aging, various systemic factors that regulate the activity and differentiation of muscle stem cells, namely satellite cells (SCs), including muscle stem cell niche factors, myogenin, and transforming growth factor beta (TGF-β), may undergo alterations, leading to a decline in their population numbers [71]. Consequently, the SCs lose their functionality, thereby compromising their ability to replace and restore damaged muscle fibers [72]. 

The pathogenesis of primary sarcopenia involves chronic inflammation and oxidative stress, primarily driven by the accumulation of Reactive Oxygen Species (ROS) due to decreased cellular antioxidant activity [73,74]. This leads to muscle atrophy and cell death, with prolonged muscle regeneration fueled by ROS-induced DNA damage and respiratory chain dysfunction, creating a self-perpetuating cycle [75,76]. Excessive anion radical production, coupled with insufficient neutralization by the superoxide dismutase (SOD) enzyme system, induces metabolic and contractile changes in muscle tissue [77]. Reduced physical activity with aging exacerbates muscle imbalance, fostering a pro-inflammatory environment that accelerates muscle deterioration [78]. Activation of atrophic factors in myotubes is influenced by levels of tumor necrosis factor alpha (TNF-α) and interleukin 6 (IL-6) in plasma and muscles, which also impair macrophage regenerative capacity [79].

### 3.2. Inflammation in IBD

The inflammatory state in the intestines of IBD patients could be the starting point for muscle depletion as it activates pathways similar to those in primary sarcopenia [35,80]. Increased levels of pro-inflammatory cytokines, causing persistent systemic inflammation, are crucial in the development of this challenging condition [80,81]. Patients sharing common features such as chronic inflammation, malnutrition, and immobility, including those with IBD, exhibit an accelerated onset of sarcopenia [4].

Tumor necrosis factor alpha has pleiotropic functions and is considered the main cause of disruption of the intestinal barrier by damaging the epithelial tight junctions (TJs), leading to an increase in gut permeability with a consequent translocation of lipopolysaccharide (LPS) into systemic circulation [82]. High levels of TNF-α are also associated with reduced muscle mass and apoptosis in muscle cells [77] through various mechanisms. For instance, the activation of the NF-kB signaling pathway leads to the expression of “atrogenes” (atrophy-related genes), which are related to muscle atrophy and promote the degradation of proteins such as muscle RING-finger protein-1 (MurF1) and Atrogin through the transcription of the E3 ligases of the ubiquitin proteasome. Moreover, TNF-α may trigger the release of nitric oxide (NO) by activating local vascular endothelial cells, leading to increased vascular permeability, which allows the passage of proinflammatory cells and triggers excessive inflammation. Bian, Ai-Lin et al. [83] focused on the relationship between sarcopenia and inflammatory factors IL-6 and TNF-α and found that serum levels of these cytokines could increase two to four times, along with an increase in C-reactive protein (CRP). In their study, increased serum levels of inflammatory cytokines and proteins were closely related to imbalanced skeletal muscle content and strength [84,85].

Interleukin 6 is often elevated in IBD and can promote skeletal muscle wasting by increasing skeletal muscle protein degradation and reducing both muscle proteosynthesis and muscle tissue anabolism. These activities seem to follow the activation of three different cellular signaling pathways, all starting from the binding of IL-6 and its receptor: the Janus kinase/signal transducer and activator of transcription (JAK/STAT) signaling pathway, the Mitogen-activated protein kinase/Extracellular signal-regulated kinases (MAPK/ERK) pathway, and the Phosphoinositide 3-kinase/Protein kinase B/mammalian target of rapamycin (PI3K/AKT/mTOR) pathway [86]. One signaling pathway in particular, the JAK/STAT pathway, is highly involved in IBD pathogenesis [87]. In IBD, IL-6 produced by macrophages and T cells is able to upregulate the production of pro-inflammatory cytokines and inhibit T cell apoptosis through IL-6 receptors, which are linked to the family of cytokine class I receptor involving Janus kinases/signal transducers and activators of transcription (JAK/STAT) pathways. Activation of JAK ends with the activator of transcription 3 (STAT3), subsequently contributing to the disruption of skeletal muscle proteosynthesis [88]. Indeed, it has been observed that a persistent increase in serum levels of this cytokine is associated with muscle atrophy and sarcopenia [83,89]. The JAK/STAT signaling pathway, in addition to being involved in the pathogenesis of IBD, appears to be one of several pathways activated by IL-6 involved in the increased protein degradation of muscle tissue [83]. Finally, IL-6 is also involved in the development of insulin resistance through various pathways, as well as TNF-α. In IBD insulin resistance condition could derive from a variety of factors like chronic exposure to IL-6, which potentiates low-grade tissue-specific and/or systemic inflammation that lastly results in insulin resistance [90,91]. It has been observed that IL-6 and TNF-α can also lead to sarcopenia establishing an insulin resistance condition since insulin is implicated in acceleration of protein synthesis and in cellular calcium uptake: insulin resistance may lead to decreased calcium uptake, which is not conducive to muscle contraction [92]. 

Lipopolysaccharide may indirectly affect the muscle in IBD. As a result of the damage and increased permeability of the intestinal barrier, which leads to translocation of LPS present on the surface of some intestinal Gram-negative bacteria into the systemic circulation. LPS can bind to toll-like receptor-2 (TLR-2) and toll-like receptor-4 (TLR-4), which are expressed by skeletal muscle fibers, resulting in an increase in the production of ROS [93]. While the precise role of mitochondrial stress in the development of IBD remains to be fully elucidated [94], existing research [95] has established the significant role of mitochondrial dysfunction in the pathogenesis of sarcopenia. The potential link between these two clinical conditions has been the subject of recent investigations, yielding some preliminary evidence to support a connection [96]. 

### 3.3. Gut Microbiota and Intestinal Permeability

Dysbiosis may contribute to the pathogenesis of IBD by disrupting the balance between protective and aggressive microbial species [97]. Moreover, disruption of the intestinal barrier is also a primary cause of IBD pathogenesis, and preclinical data support this hypothesis [98]. An abnormal immune response to microbiota and barrier disruption in genetically predisposed individuals is one of the most accepted theories behind the etiology of IBD [99]. 

Although the effect of dysbiosis and a disrupted intestinal barrier on the development of sarcopenia in IBD patients remains to be fully clarified, emerging preclinical evidence suggests that these may play a role in the pathogenesis of sarcopenia [100]. An increased abundance of genera belonging to the phylum of *Proteobacteria*, which leads to an increased translocation of LPS, has been widely documented in IBD [101,102]. These aspects have also been correlated with a loss of muscle mass.

According to a study by Bindels et al., increased intestinal permeability in cachectic mice is linked to a microbiome imbalance [103]. Further investigations in mouse models and a few studies in humans have confirmed that high intestinal permeability is associated with lower muscle mass [104,105,106,107,108,109,110]. Additionally, a number of mouse models have shown that interventions aimed at reducing intestinal permeability lead to a reduction in muscle mass loss [106,107,108,110]. 

Studies have shown that LPS, which is a component of the outer membrane of Gram-negative bacteria, can pass through a weakened gut barrier and lead to changes in skeletal muscles, similarly to the muscle phenotype of aging [111]. These findings are supported by research showing that an increased presence of Gram-negative *Proteobacteria* in the gut microbiota is significantly associated with weaker grip strength, while a positive association was found between Gram-positive *Actinobacteria* and stronger skeletal muscle [112]. Furthermore, an increase in intestinal permeability and the presence of LPS have been linked to a decline in skeletal muscle strength [112] and physical function in both healthy adults and the elderly [113].

Gut microbiota also appear to have some influence in the development of sarcopenia, through their production of metabolites and influence on gut-associated lymphoid tissue (GALT) immune cells in the production of pro-inflammatory cytokines [82]. In addition, recent studies have found that gut microbiota play a role in maintaining muscle strength [87] and physical functioning [114,115]. 

Studies indicate that older adults with higher muscle strength harbor increased levels of *Prevotella* and *Barnesiella* genera in their gut microbiome compared to those with weaker muscles. Transfer of fecal samples from these individuals to germ-free mice resulted in enhanced grip strength, particularly in mice with high-functioning colons [116]. Moreover, less frail adults exhibit a higher relative abundance of *Prevotella* and *Barnesiella* compared to frailer individuals, suggesting a potential role of these bacteria in maintaining muscle function [42,116]. Conversely, patients with inflammatory bowel disease (IBD) and frail individuals tend to exhibit reduced levels of microbial species producing short-chain fatty acids (SCFAs), such as *Faecalibacterium prausnitzii*, which plays a crucial role in IBD development [117]. 

SCFAs, but also secondary bile acids, water-soluble B vitamins, and polyphenols, are key for modulating the gut–muscle axis. They promote insulin sensitivity, amino acid biosynthesis, mitochondrial biogenesis, and muscle anabolism [118,119]. SCFAs also play a positive role in the expression of mitochondrial proteins involved in energy production, redox balance, and modulation of the inflammatory cascade [120]. Studies have shown that patients with reduced relative abundance of SCFA-producing species, like *Lactobacillus bifidobacteria*, also have decreased levels of bacteria involved in protein degradation to amino acids in the intestine, stimulation of the IGF-1/mTOR pathway, and increased expression of genes responsible for muscle protein synthesis [120,121]. Similarly, *Klebsiella* and *Escherichia coli* have been found to play a role in skeletal muscle anabolism and cell proliferation by stimulating the IGF-1/mTOR pathway [5].

It is worth noting that the microbiota can modulate amino acid bioavailability [121,122]. Research has revealed that some bacterial species can affect the entry of amino acids into the portal circulation for use throughout the body [123]. Others have the ability to facilitate the de novo synthesis of essential amino acids (EAAs), which are essential for amino acid homeostasis in the host [124]. 

### 3.4. Malnutrition

Malnutrition, as defined by the European Society for Clinical Nutrition and Metabolism (ESPEN), manifests as inadequate food intake or nutritional deficiencies, resulting in alterations in body composition and cell mass. This condition ultimately leads to impaired physical and cognitive function and potentially adverse clinical outcomes [125]. Patients with inflammatory bowel disease (IBD) frequently experience malnutrition due to chronic intestinal inflammation, resulting in significant nutritional and metabolic deficiencies and subsequent impaired nutrient absorption. The pathogenesis of malnutrition in IBD involves multifaceted factors such as reduced food intake, intestinal malabsorption, chronic protein loss in stool, and increased energy demands due to hypercatabolism [4,126]. Malnutrition in IBD appears to precede the loss of muscle mass and is attributed to the disease-induced systemic inflammatory response. Factors such as cytokines (e.g., TNF), chemokines (e.g., IL-6, IL-1), and molecules like leptin contribute to appetite loss, weight reduction, and muscle wasting, resulting in protein-energy malnutrition [127]. Despite variations in prevalence, malnutrition can occur across all disease activity levels, shortly after diagnosis, and independently of BMI, highlighting its clinical significance [128].

While much attention is given to macronutrient intake in managing IBD, the importance of micronutrients cannot be overlooked. Micronutrient deficiencies are common, influenced by disease activity, dietary intake, and supplementation. Addressing these deficiencies, particularly vitamin D and zinc, is crucial for immune function and inflammatory regulation [4]. The presence of micronutrient deficiencies within the IBD population is frequently underestimated; therefore, our attention will now be directed toward this aspect.

Vitamin D stands out among micronutrients as the sole one for which dosage recommendations are provided by the European Crohn’s and Colitis Organisation (ECCO) and the European Society of Gastrointestinal and Abdominal Radiology for all patients with IBD [129]. Vitamin D has multiple and pleiotropic functions; however, interestingly animal studies suggest that vitamin D deficiency exacerbates intestinal inflammation by activating inflammatory signaling pathways, and this is one of the mechanisms that may contribute to sarcopenia [5]. 

Zinc deficiency has garnered attention for its potential role in exacerbating IBD pathogenesis and disease course [126,130,131]. Furthermore, zinc plays a crucial role in maintaining epithelial barrier integrity and immune cell function [132]. A recent metanalysis showed evidence of the relationship between zinc deficiency and sarcopenia [133].

Other micronutrient deficiencies, including vitamins A, K, C, B6, and B1, as well as selenium, have been associated with sarcopenia in numerous studies [134,135,136]. These deficiencies are commonly observed in patients with IBD despite the absence of specific studies conducted on this population. Nevertheless, these micronutrients play diverse immunomodulatory roles and are essential for various physiological processes, including antioxidant defense and tissue repair. Therefore, their deficiency is likely linked to sarcopenia in individuals with IBD. Potential pathogenetic mechanisms that can cause sarcopenia in IBD are illustrated in Figure 1.

## 4. Impact of Sarcopenia on IBD Clinical Settings

### 4.1. Outcomes in IBD

The impact of malnutrition and sarcopenia in IBD patients is well described in terms of a more severe disease course, poor response to biological agents [137], poor outcomes after surgery [19], and longer hospital stays [35]. A recent systematic review reported that 42% of IBD patients undergoing surgery had sarcopenia when assessed radiologically [4]. The correlation between sarcopenia and perioperative mortality has been demonstrated for different types of surgeries [138,139,140,141] and is relevant for the management of IBD patients who are mostly young, deeply malnourished, immunosuppressed, and often require urgent surgical resolution for disease flare ups. Liu et al. [142] proposed a study in which they evaluated the impact of sarcopenia, diagnosed according to the Asian Working Group for Sarcopenia 2019 (AWGS2019) criteria, on clinical outcomes in patients with IBD. They highlighted that sarcopenia is associated with poor outcomes after surgery complications, longer hospital stays, and, subsequently, higher costs [143], altered body composition, and a reduced QoL. Given the above, muscle function assessment could play a crucial role in preventing sarcopenia as an early predictor of it, as function deteriorates more rapidly than muscle mass. In IBD, along with the elevation of inflammatory direct indicators like CRP and Erythrocyte Sedimentation Rate (ESR), which reflect the severe inflammatory response [5], researchers identified lower BMI and albumin serum levels as other risk factors for sarcopenia, which are directly connected to the patient’s nutritional status [128,144]. These factors may be detectable, making early screening, diagnosis, and intervention to improve the prognosis in IBD patients feasible. A recent literature overview suggested sarcopenia as an independent predictor of poor postoperative outcome after major surgery, and the positive association has been evaluated in multiple retrospective cohort studies [143,145,146]. The meta-analysis conducted by Erös et al., which included ten studies involving a total of 885 IBD patients, revealed sarcopenia as an independent predictor of overall postoperative complications [147]. However, no discernible difference was identified when specific complications were analyzed individually. Knoedler et al. [148] delved into specific peri-operative complications among sarcopenic patients undergoing colorectal resection for IBD, juxtaposing outcomes with those of patients exhibiting normal muscle mass. Their findings aligned with the literature trend, indicating heightened postoperative and infectious complications in sarcopenic individuals. Additionally, they established sarcopenia as an independent predictor for anastomotic leak following colorectal surgery. These results suggest that radiological assessment of sarcopenia should be taken in consideration in the multi-factorial decision to perform a primary anastomosis or diverting stoma for patients undergoing colorectal surgery for IBD. Ge et al. [149] investigated the predictive value of sarcopenia in the clinical course and postoperative outcomes of acute severe ulcerative colitis (ASUC). According to the Oxford regimen, for ASUC patients who do not respond to the first-line therapy with i.v. steroids, switching to medical rescue therapy should be advised [150,151]. If rescue therapy fails, surgery should be performed; indeed, although medical treatments progresses, approximately 20–40% of UC patients undergo proctocolectomy with ileal pouch–anal anastomosis (IPAA), which is the most commonly performed surgery [152,153]. In their study, Ge et al. evidenced a high prevalence of sarcopenia in patients with ASUC, which was also identified as an independent risk factor for intravenous steroid infusion (IVS) and refractory and medical rescue therapy failure. In addition, they highlighted that sarcopenia was an independent biomarker for postoperative complications in ASUC, defined as those that occurred before hospital discharge or within 30 days after surgery and classified based on the Clavien–Dindo system [152], in line with recent meta-analysis findings.

In conclusion, sarcopenia has been demonstrated in multiple studies to be an independent risk factor for surgery and postoperative complications, such as postoperative abdomino-pelvic sepsis, surgical site infections, and thrombotic complications, including greater and prolonged hospitalizations, hospital re-admission, and mortality, in patients with IBD [6,7,19]. The clinical outcomes from the principal studies on sarcopenia in IBD are summarized in Table 2.

### 4.2. Role of IBD Medications

In addition to well-established factors contributing to the loss of response to biological therapies, such as the production of antidrug antibodies leading to accelerated monoclonal antibody (mAb) clearance, other factors, including gender, body size, disease type, the degree of systemic inflammation, malnutrition, and serum albumin concentration may also play a role in this mechanism [155]. It has been demonstrated that IBD patients with hypoalbuminemia exhibit lower infliximab response rates and experience an acceleration in infliximab clearance. Similarly, sarcopenia has been associated with diminished response to biologic therapies. For instance, Ding et al. conducted a study indicating that myopenia was linked to primary nonresponse to anti-TNF in 106 patients with moderate to severe CD [156]. Grova et al. [154] investigated the role of sarcopenia in predicting clinical and endoscopic outcomes in patients with Crohn’s disease (CD), highlighting sarcopenia as a negative prognostic factor for endoscopic remission (ER) in CD patients treated with biologics. Another effort to assess the association between sarcopenia and biological agents in CD patients was undertaken by Liu et al. [10]. The study, which included 94 CD patients, revealed that sarcopenia serves as an independent risk factor for primary loss of response (LOR). Moreover, the researchers developed a predictive model for LOR to biological agents based on sarcopenia, Montreal L1 type, presence of perianal lesions, and serum monocyte percentage available at baseline. They elucidated that the identification of sarcopenia through CT or MRI prior to administering biologics correlates with LOR to biological agents or infliximab in adult CD patients, alongside factors such as perianal lesions, a higher serum monocyte percentage, and the isolated ileum type. A recent meta-analysis conducted by Patsalos et al. [157] focused on the effects of TNF-a inhibitor therapy on body weight and BMI, evidencing a significant increase in BMI but only a small increase in body weight in adults with immune-mediated inflammatory diseases, including IBD, after TNF-a inhibitor initiation, suggesting that weight gain during anti-TNF-a therapy may be due to a restoration of normal body weight.

Hence, medications commonly used in IBD, in particular anti-TNF-α agents, are useful agents to target not only the inflammatory bowel disease but also the associated sarcopenia due to shared pathogenetic pathways. Indeed, Subramaniam et al. demonstrated that anti-TNF-α agents rapidly reversed sarcopenia and increased muscle volume and strength by targeting inflammatory mediators involved in inflammatory pathways shared by both IDB and sarcopenia [158]. Santos et al.’s [159] findings also corroborate an observation by Subramaniam et al. which associates anti-TNF therapy with increases in body composition parameters related to lean and fat mass. Anti-TNF-α agents are commonly used to treat IBD because they are effective in induction and maintenance of response and remission of disease [160]. Since anti-TNF-α agents like infliximab reduce levels of nuclear NF-κB in targeted organs, they demonstrated improvement in both conditions, plus a demonstrated improvement also in osteopenia [161,162].

To date, limited data are available concerning other commonly used drugs for IBD, such as ustekinumab, vedolizumab, and tofacitinib [161,163]. Regarding anti-JAKs, as these small-molecule drugs disrupt the Janus kinase signaling pathway, which includes IL-6 involved in its pro-inflammatory functions, they could theoretically improve nutritional status and reduce muscle wasting [164]. Nonetheless, future studies should investigate the impact of these drugs on sarcopenia in patients with IBD. Table 3 summarizes current studies investigating the effects of IBD medications on sarcopenia.

### 4.3. Sarcopenic Obesity

To date, 15% to 40% of individuals with IBD are obese [165,166], defined by a BMI > 30 kg/m^2^ [167], while 5% of individuals meet the criteria of sarcopenic obesity [168]. Research suggests that IBD patients may have an increased risk of developing obesity and sarcopenic obesity. A recent study found that the percentage of patients with obesity increased from 23% to 31% within 24 months after diagnosis, associated with a decrease in the appendicular skeletal muscle mass index [169]. The reason is still partly unknown, but certainly, the use of steroid therapy and lack of physical activity can lead to significant changes in body weight and promote sarcopenic obesity in IBD patients [163]. Obesity is known to be associated with a more severe IBD clinical course [170] and an increased risk of postoperative complications [166,171]. Patients with obesity and IBD generally respond even less well to advanced treatments. In fact, pharmacokinetics and the ongoing inflammatory effects of obesity can interfere with the effectiveness of biologic treatments [163,166]. Obesity is associated with an increase in the production of pro-inflammatory adipokines, which are produced by adipose tissue and are involved in various physiological functions, including the immune response [171,172]. These hormones may play a role in maintaining the activity of inflammatory bowel disease (IBD). Leptin, in particular, appears to play a significant role in IBD by participating in various molecular pathways associated with inflammation. Recent research suggests that leptin influences the regulation of CD4+ CD25+ regulatory T-cell proliferation, which could explain its involvement in the autoimmune disease process [173]. Among effector T cells, leptin promotes orientation towards Th1 and the production of pro-inflammatory cytokines such as TNF-α, IL-2, IL-6 [174], IL-18, and plasminogen activator inhibitor (PAI-1) [79,175].

An experimental model of colitis showed that administering leptin antagonists promoted the recruitment of T-reg cells while decreasing the infiltration of pro-inflammatory cytokines into the colon [168,176].

IBD patients are prone to sarcopenia due to the processes related to the disease itself and to all the mechanisms previously discussed. Moreover, obese sarcopenic patients may also experience anabolic resistance, which occurs when fat accumulates within the muscles. This resistance causes the body to become unresponsive to stimuli such as growth factors, hormones, amino acids, and exercise, which ultimately leads to a decrease in muscle protein synthesis [177] and thus an increased risk of sarcopenic obesity [178]. Aspects related to sarcopenic obesity, including mechanisms, outcomes, and potential treatments are summarized in Figure 2.

## 5. Treatments

Insufficient oral food intake is a major cause of malnutrition in individuals with inflammatory bowel disease (IBD). This is primarily due to self-imposed food limitations for fear of experiencing symptoms and the presence of symptoms such as nausea, abdominal pain, vomiting, and diarrhea. Furthermore, individuals with IBD often follow an altered diet that lacks adequate nutritional content, which can lead to deficiencies [179].

To counteract sarcopenia, optimization of nutritional status is an effective countermeasure [180] in improving clinical and postoperative outcomes [4,181]. In cases of reduced dietary intake, enteral nutrition (EN) can help maintain or restore nutritional status by providing beneficial effects on protein turnover [182,183]. When patients are unable to feed themselves orally or enterally, parenteral nutrition is recommended [183].

Currently, there is no established evidence to support the role of specific dietary patterns in preventing or treating sarcopenia in IBD [5]. However, it is well known that protein intake plays a crucial role in preventing and treating sarcopenia. In IBD patients, ESPEN guidelines recommend a daily protein intake of 1 g/kg/day in case of remission and 1.2–1.5 g/kg/day in case of active disease [183]. Nonetheless, for effective management of sarcopenia, it is essential to also consider protein quality [184]. Protein quality can be assessed by its digestible indispensable amino acid score (DIAAS), a measure of the quality of protein based on the content of indispensable amino acids and their digestibility [185].

### 5.1. Supplements

Amino acids intervene in stimulating protein synthesis and preventing muscle protein degradation [186]. Several studies have demonstrated the effectiveness of essential amino acid (EAA) supplementation alone in elderly people with sarcopenia in increasing lean body mass [187,188,189]. Further studies found no significant results following protein or amino acid supplementation in patients with sarcopenic obesity [190,191] without concomitant nutritional or exercise intervention.

Leucine, among amino acids, has garnered significant attention in numerous studies. A systematic review and meta-analysis have highlighted that the ingestion of leucine significantly increases the rate of fractional synthesis of muscle proteins in older individuals [192,193,194,195].

β-hydroxy β-methylbutyric acid (HMB), a metabolite of leucine [196], appears to be a valuable ally in promoting muscle protein synthesis and inhibiting muscle protein breakdown [196,197,198,199,200]. HMB supplementation has been shown to be effective in maintaining muscle mass in patients during a 10-day rest period, where the control group showed a significant decrease in total lean body mass [201]. However, there are currently no specific studies on IBD patients.

Vitamin D supplementation improves muscle strength in adults [202]. When combined with exercise and protein supplementation, it can also significantly improve grip strength [203]. These results are not surprising, considering that vitamin D suppresses the expression of myostatin in muscle tissue, which is an inhibitor of muscle growth [204] thus theoretically leading to muscle synthesis. Moreover, vitamin D also plays an important role in the metabolism of calcium and inorganic phosphate, both of which participate in muscle contractility [205]. In IBD, limited studies have identified the effect of vitamin D supplementation on muscle mass or strength [5], despite the demonstrated benefits of vitamin D supplementation in the disease itself [206].

Omega-3 polyunsaturated fatty acids are involved in various processes, including reducing pro-inflammatory cytokines that contribute to catabolism [207,208] and activating the mTORC1 pathway, which increases muscle protein synthesis (MPS) and amino acid transport [209,210]. In preclinical and clinical studies, an inverse association between polyunsaturated fatty acids (PUFAs) and sarcopenia has been demonstrated [211,212,213]. It appears that omega-3 supplementation can improve muscle mass, strength, and physical performance [209], even without a concomitant training program [214]. Although a significant amount of evidence supports the importance of implementing additional interventions to prevent muscle loss and maintain strength and function, only a limited number of studies have examined this effect in IBD patients specifically [215], and more research is needed to better address the management of sarcopenia in these patients.

### 5.2. Resistance Training

Regular exercise has already been shown to improve body composition and control inflammation by promoting an anti-inflammatory state [216]. A distinction can be made between endurance training (ET), which involves low-resistance work for prolonged periods of time, and resistance training (RT), characterized by more powerful movements of shorter duration [217,218]. In chronic conditions such as IBD, there may be an imbalance between the processes of proteolysis and muscle protein synthesis. Previous studies showed that engaging in resistance exercises can boost the muscle protein synthesis rate, which leads to a rise in the muscle amino acid balance for up to 48 h following the workout [219,220]. The exact mechanisms behind this process are not yet fully understood, but it appears that IGF-1 plays a fundamental role. RT can lead to an increase in IGF-1 secretion by skeletal muscle, which in turn promotes the hypertrophy of muscle fibers, enhances muscle strength, and prevents muscle loss [221,222,223]. According to recent studies, patients with Crohn’s disease who practiced RT showed significant improvements in muscle strength, muscle function, and bone mineral density [224,225].

In patients with IBD and sarcopenic obesity, the goal of treatment is to reduce body weight while preserving muscle mass to reduce complications and increase the response to biological treatments [165]. Current guidelines suggest that these patients should have an increased protein intake of 1.2–1.5 g/kg during both the active and remission phases of the disease [165,226].

The efficacy of dietary intervention as a standalone treatment for increasing muscle mass in patients remains uncertain [227]. However, exercise is believed to play a significant role in patients with sarcopenic obesity. Indeed, on one hand, it contributes to increased energy expenditure, and on the other hand, it is the main anabolic stimulus leading to muscle protein synthesis [228,229]. The literature shows the effectiveness of both resistance training [230], aerobic training [231], as well as simultaneous exercise [232]. Consequently, the combination of a low-calorie, high-protein diet and physical training appears to be the best strategy. In severe cases, obesity pharmacotherapy and bariatric surgery should be considered as additional therapies for obese IBD patients [165]. We assert that effective management of sarcopenia necessitates a multidisciplinary approach. This strategy should encompass the collaboration of diverse health professionals, including a dedicated dietitian who can provide ongoing and sustained nutritional support and exercise professionals who can design tailored exercise programs. All these measures, which have not yet been incorporated into daily clinical practice, would lead to a global improvement in patient outcomes, particularly regarding disease activity, response to pharmacological, and surgical treatments, and thus enhance their QoL. Further research is needed to integrate various studies and establish novel management strategies for sarcopenia. Figure 3 summarizes potential treatments for sarcopenia in IBD.

## 6. Conclusions

Integrating comprehensive assessments of muscle mass, strength, and function into routine clinical practice can facilitate timely intervention and personalized treatment plans tailored to individual patient needs, thus improving clinical outcomes. Several methods are available to assess sarcopenia; MRI and CT are commonly used in clinical practice and can provide valuable information about sarcopenia. DXA and BIA are also reliable methods, although ultrasound is gaining popularity as a reliable, inexpensive, and widely available diagnostic method. Nutritional interventions, including enteral and/or parenteral nutrition, protein supplementation, and amino acid interventions, play a vital role in preserving muscle mass and function while addressing the malnutrition commonly observed in IBD patients. Moreover, emerging evidence suggests the potential benefits of gut microbiota modulation, exercise interventions, and pharmacotherapy in mitigating muscle loss and improving physical function in IBD-related sarcopenia. Future research efforts should focus on elucidating the specific mechanisms underlying the gut–muscle axis, optimizing therapeutic strategies, and evaluating the long-term efficacy and safety of interventions in diverse patient populations.

In summary, the bidirectional relationship between IBD and sarcopenia represents a significant clinical challenge with profound implications for patient outcomes and quality of life. Comprehensive assessment, early detection, and targeted interventions addressing inflammation, malnutrition, and muscle wasting are essential components of holistic management strategies for individuals with IBD-related sarcopenia. Collaborative efforts between clinicians, researchers, and allied health professionals are paramount in achieving these goals and addressing the unmet needs of this vulnerable patient population. Future research should enhance our comprehension of the biological mechanisms that underlie the relationship between the gut and muscle in order to develop targeted therapies that can modulate gut microbiota, muscle protein synthesis, and inflammatory processes to maintain or improve muscle mass and functionality.

## Figures and Tables

**Figure 1 biomedicines-12-01218-f001:**
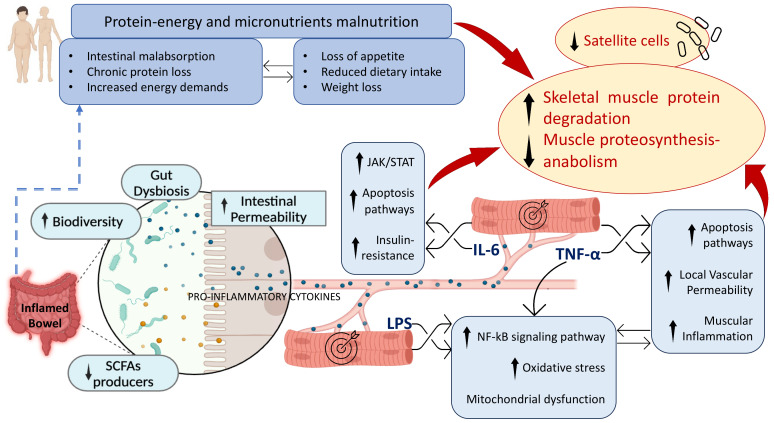
Pathogenetic mechanisms of sarcopenia in IBD. SCFAs: short-chain fatty acids; LPSs: lipopolysaccharides.

**Figure 2 biomedicines-12-01218-f002:**
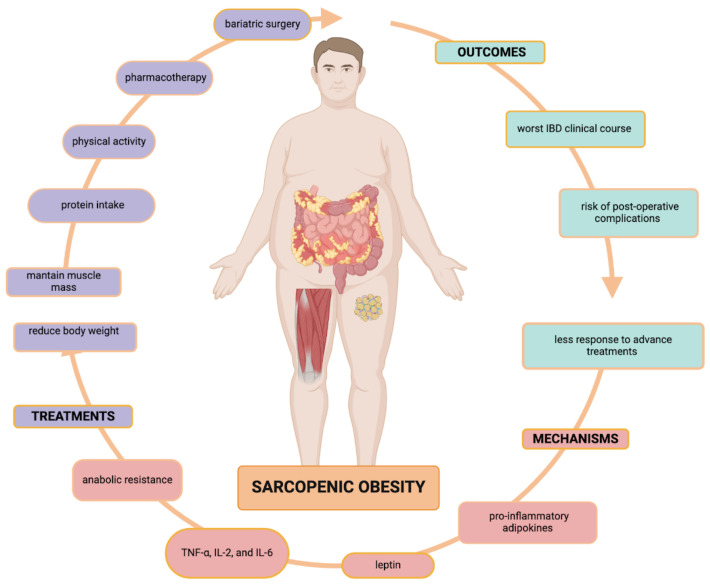
Sarcopenic obesity in IBD.

**Figure 3 biomedicines-12-01218-f003:**
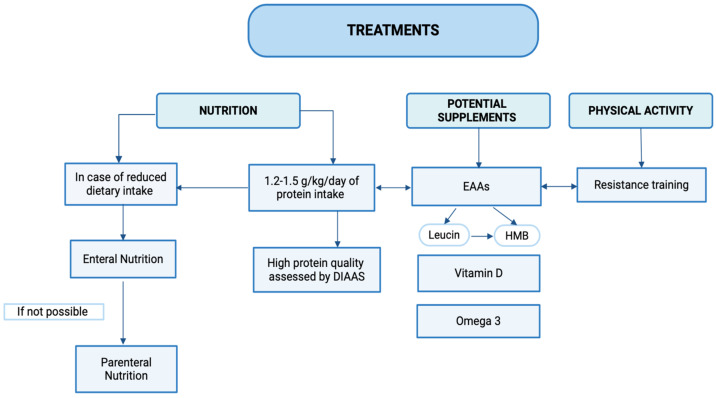
Potential treatments for sarcopenia in IBD. DIAAS: digestible indispensable amino acid score; EAAs: essential amino acids; HMB: β-Hydroxy β-methylbutyric acid.

**Table 1 biomedicines-12-01218-t001:** Diagnostic methods to assess muscle mass.

	DIAGNOSTIC METHODS			
	Assessment	Advantages	Limits	IBD
CT [22,23,24,25]	CSA Myosteatosis Myofibrosis	High precision Reliable outcomes	Ionizing radiation High costs	Commonly used in IBD clinical practice
MRI [29,30,31,32,34]	CSA Muscle quality Myofibrosis Myosteatosis	Non-exposure to radiation Accurate for body composition	High costs Long image capture time Limited availability	Commonly used in IBD clinical practice
US [21,43,44,45,46,47,48,49]	CSA Muscle thickness Myofibrosis Myosteatosis	Accurate Easily repeatable Safe Low-cost	Lack of standardized cut-offs Operator- dependent	Commonly used in IBD clinical practice Influenced by inflammation
DXA [38,39,40,41,42]	Visceral tissue Subcutaneous adipose tissue Total lean mass	Non-invasive Low radiation exposure Low-cost	Indirect assessment Influenced by hydration status	Valid tool for analyzing body composition
BIA [20,35,36,37,38]	Fat mass Muscle mass Membrane integrity Cellularity	Rapid Portable Non-invasive Low-cost	Indirect assessment Influenced by hydration status	Reliability in assessing body composition Influenced by inflammation

CT: computed tomography; MRI: magnetic resonance imaging; US: ultrasonography; DXA: dual-energy X-ray absorptiometry; BIA: bioelectrical impedance analysis; CSA: cross-sectional area.

**Table 2 biomedicines-12-01218-t002:** Sarcopenia and clinical outcomes.

SARCOPENIA AND CLINICAL OUTCOMES			
Author, (Year), Study Type	Outcomes	Patients (*n*)	Results
Erös et al. (2020), systematic review [147]	Sarcopenia on surgical outcomes	885	Need for surgery (OR: 2.665; 95% CI 1.121–6.336; *p* = 0.027); postoperative complications OR = 6.097; 95% CI 1.756–21.175; *p* = 0.004
Liu et al. (2022), prospective cohort study [142]	Sarcopenia on clinical outcomes	110	Rates of surgery (OR = 6.651; 95% CI: 2.333–18.959; *p* < 0.001); re-hospitalization (OR = 6.344; 95% CI: 2.874–14.003; *p* < 0.001); death, *p* = 0.003
Knoedler et al. (2023), systematic review [148]	Sarcopenia on surgical outcomes	97,643	Surgical complications (OR 1.92; 95% CI:1.20–3.07; *p* = 0.007)
Ge et al. (2021), retrospective cohort study [149]	Sarcopenia on clinical course of ASUC	233	Colectomy (OR = 3.411; 95% CI, 1.147–10.141; *p* = 0.027); postoperative complications (OR = 4.157; 95% CI: 1.364–12.667; *p* = 0.012)
Grova et al. (2023), retrospective observational study [154]	Sarcopenia on endoscopic remission	358	Endoscopic remission (OR = 5.2; 95% CI 1.60–16.8; *p* = 0.006)

ASUC = acute severe ulcerative colitis.

**Table 3 biomedicines-12-01218-t003:** Sarcopenia and IBD medications.

SARCOPENIA AND IBD MEDICATIONS				
Author, (Year), Study Type	Outcomes	Patients (*n*)	Medication	Results
Santos et al. (2017), prospective cohort study [159]	Anti-TNF-α therapy on BC	23	Anti-TNF-α	↑ LM *p* < 0.0001 ↑ FM *p* < 0.0001
Liu et al. (2023), retrospective cohort study [10]	Sarcopenia and LOR to biologic agents	94	Anti-TNF-α	Primary LOR (OR = 2.87, 95% CI: 1.07–7.69)
Subramaniam et al. (2015), prospective cohort study [158]	Anti-TNF-α on muscle volume and strength	19	Anti-TNF-α	↑ muscle volume (*p* = 0.010) ↑ muscle strength (*p* = 0.002)
Ding et al. (2017), systematic review [156]	BC on anti-TNF-α therapy response	106	Anti-TNF-α	Primary non-response (OR = 2.93; CI: 1.28–6.71, *p* = 0.01)
Patsalos et al. (2020), systematic review [157]	Anti-TNF-a on body weight and BMI	1245	Anti-TNF α	↑ body weight (SMCC = 0.23, 95% CI 0.10–0.37; *p* = 0.0006) ↑ BMI (SMCC = 0.26, 95% CI 0.13–0.39; *p* < 0.0001)

BMI = body mass index; FM = fat mass; IBD = inflammatory bowel disease; LOR = loss of response; LM = lean mass; SMCC = standardized mean change; TNF-α = tumor necrosis factor α.

## Data Availability

No new data were created or analyzed in this study. Data sharing is not applicable to this article.

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
