# Peer review of "Navigating the Intersection: Sarcopenia and Sarcopenic Obesity in Inflammatory Bowel Disease"

_biomedicines, 2024, doi:10.3390/biomedicines12061218_

Round 1
Reviewer 1 Report
Comments and Suggestions for Authors
1. Only three key words are not enough.
2. For table 1, the authors failed to present which literatures they cited.
3. For section 3, a figure about the detailed mechanisms should be presented.
4. For the inflammation of IBD, the authors failed to connect the cytokines, signaling pathways and related immune cells.
5. Figure 1 is too superficial without detailed mechanisms.
6. Please provide tables of in vitro and in vivo studies.
7. For the clinical studies, please supplement detailed mechanisms in tables.
8. Where is Figure 2?
Comments on the Quality of English LanguageThe manuscript need extensive revision in language.
Author Response
1. Only three key words are not enough.
Thank you for your recommendation. We have increased the number of keywords.
2. For table 1, the authors failed to present which literatures they cited.
Thank you for your feedback. We have taken your suggestion into account and incorporated the references into Table 1.
3. For section 3, a figure about the detailed mechanisms should be presented.
We have provided an image illustrating the mechanisms. For further in-depth information, please consult the accompanying text.
4. For the inflammation of IBD, the authors failed to connect the cytokines, signaling pathways and related immune cells.
We thank you for your comment. However, although the topic was briefly addressed in our review, it was not a primary focus of our paper. The intent of this paragraph was to provide a succinct explanation of inflammation in IBD in preparation for the subsequent discussion on sarcopenia in IBD.
5. Figure 1 is too superficial without detailed mechanisms.
We have provided an image illustrating the mechanisms. For further in-depth information, please consult the accompanying text.
6. Please provide tables of in vitro and in vivo studies.
Thank you for the suggestion. However, section 3 already contains a figure, and the document as a whole includes three tables. Furthermore, the studies cited are integrated into the text and explained. As a result, we do not consider it necessary to include it. If, in your opinion, it is strictly necessary, we will include it in a future revision.
7. For the clinical studies, please supplement detailed mechanisms in tables.
We have included an image to illustrate the mechanisms. Please refer to the accompanying text for more detailed information.
8. Where is Figure 2?
Figure 2 is in the text.
Reviewer 2 Report
Comments and Suggestions for Authors
The current manuscript covers a relevant and comprehensive topic, addressing the intersection of sarcopenia, sarcopenic obesity, and IBD. It highlights diagnostic methods, pathophysiological mechanisms, clinical implications, and management strategies. However, there are some gaps in the manuscript that could be improved by the authors.
Comments:
Lines 34-38: This paragraph cannot be understood by a reader as the beginning of a sentence. Please rewrite again.
Line 41: Please check the current numbers "34816624"!!
Line 43: Refrence 3 should be up to date.
Line 61: "as multiple studies have conclusively established" Please change and rewrite to" according to a recent study by Wei et al 2023".
Because you mentioned just a one reference at the end of text which was "Sarcopenic Obesity: Epidemiology,...", so the "multiple studies" at the text is wrong and not supported it.
Overall, the introduction is well-structured and covers the essential aspects related to sarcopenia in IBD. It effectively introduces the topic, establishes its significance, and outlines the scope of the review. However, it would be beneficial to provide more specific references for the statements made.
Line 170: At table 1 the authors have good difinated and explained about various diagnostic methods are available to assess skeletal muscle. I recommend to authors write "Diagnostic methods" as title at the first line at the table. Also, The authors should be write all of abbreviation again at the end of the table.
General, it's important to note that no single diagnostic method can capture the complete picture of sarcopenia. Therefore, a combination of assessments, including both imaging techniques and functional tests, is often recommended for a comprehensive evaluation of sarcopenia in IBD patients. The choice of diagnostic methods may depend on factors such as availability, cost, and the specific research or clinical setting.
Lines 182-187: The authors mentioned that "There haven't been any studies conducted so far that have included this test specifically for patients with inflammatory bowel disease (IBD)." Could you please explain the reason for this statement and rewrite it?
Lines 188-200: Could you write and explain about the gaps found in each questionnaire?
Lines 320 and 326: The reference for "Studies" is not supported according to your text. You should change the word or add relevant references.
Lines 331, 348, 355, 359: All bacteria should be italicized. Please review your text for this.
Line 642: The resolution of Figure 3 appears to be unclear. Please increase the quality.
Lines 644-667: Some sentences in your conclusion are repeated. It's better to delete them. Also, refer to the gaps found in your review to help future researchers with their studies
Comments on the Quality of English LanguageMinor editing of English language required.
Author Response
Lines 34-38: This paragraph cannot be understood by a reader as the beginning of a sentence. Please rewrite again.
Thank you for your suggestion. We have revised the sentence in question.
Line 41: Please check the current numbers "34816624"!!
We have replaced the reference.
Line 43: Reference 3 should be up to date.
We have replaced the reference.
Line 61: "as multiple studies have conclusively established" Please change and rewrite to" according to a recent study by Wei et al 2023". Because you mentioned just a one reference at the end of text which was "Sarcopenic Obesity: Epidemiology,...", so the "multiple studies" at the text is wrong and not supported it.
We have rewritten according to your suggestion.
Line 170: At table 1 the authors have good difinated and explained about various diagnostic methods are available to assess skeletal muscle. I recommend to authors write "Diagnostic methods" as title at the first line at the table. Also, The authors should be write all of abbreviation again at the end of the table.
We have inserted your suggested title and re-written all abbreviations at the end of the table.
Lines 182-187: The authors mentioned that "There haven't been any studies conducted so far that have included this test specifically for patients with inflammatory bowel disease (IBD)." Could you please explain the reason for this statement and rewrite it?
Thank you for your suggestion. We have better explained the concept in the text.
Lines 188-200: Could you write and explain about the gaps found in each questionnaire?
Thank you for your recommendation. We have provided additional clarification regarding the identified gaps in each questionnaire.
Lines 320 and 326: The reference for "Studies" is not supported according to your text. You should change the word or add relevant references.
Thank you for your recommendation. We provided to change the word in question.
Lines 331, 348, 355, 359: All bacteria should be italicized. Please review your text for this.
We have italicized all instances of bacteria in the text.
Line 642: The resolution of Figure 3 appears to be unclear. Please increase the quality.
We have increased the quality of Figure 3.
Lines 644-667: Some sentences in your conclusion are repeated. It's better to delete them. Also, refer to the gaps found in your review to help future researchers with their studies.
Thank you for the observation, we have refined the conclusion by eliminating redundancies.
Reviewer 3 Report
Comments and Suggestions for Authors
The review is very interesting as it provides a comprehensive overview of sarcopenia in inflammatory bowel disease (IBD), elucidating diagnostic methodologies, pathophysiological mechanisms, clinical implications, and management. Overall, this review covers various aspects of myopathy in IBD including diagnostic methods, pathophysiological mechanisms, clinical implications, management, and treatment. The article follows a clear structure that begins with an overview of the prevalence and importance of myopathy in IBD before delving into its diagnostic methods, etiology, and potential interventions. This logical progression allows readers to systematically understand different aspects of myopathy in IBD. Additionally, the language used in this article is accurate without any noticeable linguistic issues which makes it highly readable.
Here are some comments:
1. In the Introduction, authors can further emphasize the severity of sarcopenia in IBD patients and the urgency to study it in order to attract readers' interest.
2. In the Treatment section, authors can explore potential therapeutic targets or emerging treatment methods in addition to existing interventions, providing direction for future research.
Author Response
- In the Introduction, authors can further emphasize the severity of sarcopenia in IBD patients and the urgency to study it in order to attract readers' interest.
Thank you for your indication, we have made the necessary modifications accordingly.
- In the Treatment section, authors can explore potential therapeutic targets or emerging treatment methods in addition to existing interventions, providing direction for future research.
We have made the necessary modifications as per your indication. Thank you for bringing it to our attention.